# Gasdermin D Inhibitor Necrosulfonamide Alleviates Angiotensin II-Induced Abdominal Aortic Aneurysms in Apolipoprotein E-Deficient Mice

**DOI:** 10.3390/biom14060726

**Published:** 2024-06-19

**Authors:** Jia Guo, Qing Zhang, Zhidong Li, Min Qin, Jinyun Shi, Yan Wang, Wenjia Ai, Junjie Ju, Makoto Samura, Philip S Tsao, Baohui Xu

**Affiliations:** 1Department of Cardiovascular Medicine, First Hospital Shanxi Medical University, Taiyuan 030001, Shanxi, China; zhangqing55@sxmu.edu.cn (Q.Z.); qinmin91077@163.com (M.Q.); shijy0306@126.com (J.S.); 2Department of Surgery, Stanford University School of Medicine, Stanford, CA 94305, USA; wenjiaai@stanford.edu (W.A.); jujunjie@stanford.edu (J.J.); samura@stanford.edu (M.S.); baohuixu@stanford.edu (B.X.); 3Department of Pharmacology, School of Basic Medicine, Shanxi Medical University, Taiyuan 030001, Shanxi, China; 4Institute of Medical Innovation and Research, Peking University Third Hospital, Beijing 100191, China; yanwang2019@bjmu.edu.cn; 5Cardiovascular Institute, Stanford University School of Medicine, Stanford, CA 94305, USA; ptsao@stanford.edu; 6VA Palo Alto Health Care System, Palo Alto, CA 94304, USA

**Keywords:** abdominal aortic aneurysms, gasdermin D, necrosulfonamide

## Abstract

Abdominal aortic aneurysm (AAA) is a chronic aortic disease that lacks effective pharmacological therapies. This study was performed to determine the influence of treatment with the gasdermin D inhibitor necrosulfonamide on experimental AAAs. AAAs were induced in male apolipoprotein E-deficient mice by subcutaneous angiotensin II infusion (1000 ng/kg body weight/min), with daily administration of necrosulfonamide (5 mg/kg body weight) or vehicle starting 3 days prior to angiotensin II infusion for 30 days. Necrosulfonamide treatment remarkably suppressed AAA enlargement, as indicated by reduced suprarenal maximal external diameter and surface area, and lowered the incidence and reduced the severity of experimental AAAs. Histologically, necrosulfonamide treatment attenuated medial elastin breaks, smooth muscle cell depletion, and aortic wall collagen deposition. Macrophages, CD4^+^ T cells, CD8^+^ T cells, and neovessels were reduced in the aneurysmal aortas of necrosulfonamide- as compared to vehicle-treated angiotensin II-infused mice. Atherosclerosis and intimal macrophages were also substantially reduced in suprarenal aortas from angiotensin II-infused mice following necrosulfonamide treatment. Additionally, the levels of serum interleukin-1β and interleukin-18 were significantly lower in necrosulfonamide- than in vehicle-treated mice without affecting body weight gain, lipid levels, or blood pressure. Our findings indicate that necrosulfonamide reduced experimental AAAs by preserving aortic structural integrity as well as reducing mural leukocyte accumulation, neovessel formation, and systemic levels of interleukin-1β and interleukin-18. Thus, pharmacologically inhibiting gasdermin D activity may lead to the establishment of nonsurgical therapies for clinical AAA disease.

## 1. Introduction

Abdominal aortic aneurysm (AAA) is a life-threatening degenerative aortic disease characterized by the progressive dilation of aortas, particularly the infrarenal aorta [1]. As the disease progresses, it may lead to a fatal rupture with severe morbidity and mortality [2]. Open surgery and endovascular aortic repair are the only treatments for patients meeting surgical repair criteria; no pharmacological therapies have been proven effective in treating clinical AAA disease [2,3,4,5].

Accumulating evidence has suggested that various inflammatory mechanisms are involved in the initiation and progression of AAA disease [5,6,7,8,9,10,11,12,13,14,15,16]. Inflammasomes, as represented by the well-characterized nucleotide-binding domain, leucine-rich-containing family, and pyrin domain-containing-3 (NLRP3), are large protein complexes in which activated caspase 1 cleaves interleukin (IL)-1β and IL-18 to become biologically active [17,18,19,20]. Caspase 1 also cleaves gasdermin D (GSDMD), a downstream molecule of the NLRP3 inflammasome, to C- and N-terminals (p30 fragment), with the latter forming a cell membrane pore via oligomerization for cytokine release and even pyroptosis [21,22]. Recently, GSDMD has been shown to mediate the adhesion and rolling of neutrophils in vessels in situ, the steps required for neutrophil migration [23]. It has also been reported that the cleaved GSDMD N-terminal was elevated in clinical and experimental aneurysmal aortas [24,25,26]. Necrosulfonamide (NSA) is a newly identified GSDMD inhibitor [27,28,29], which directly binds GSDMD to prevent its N-terminal oligomerization, thus blocking pore formation and ultimately cytokine release and pyroptosis [27]. Thus, we hypothesized that NSA treatment may attenuate the formation and progression of experimental AAAs.

Chronic subcutaneous infusion of angiotensin (Ang) II induces AAAs in the suprarenal aorta in male hyperlipidemic apolipoprotein (ApoE)-deficient (ApoE^−/−^) mice in a blood pressure-independent manner [30,31]. AAAs induced by Ang II infusion simulate several features of clinical AAAs, such as the progressive expansion of the aortic lumen due to the destruction of medial elastin and smooth muscle cells (SMCs), extracellular matrix remodeling, and the accumulation of macrophages, B cells and T lymphocytes [32]. In this study, we utilized an Ang II-induced experimental AAA model in ApoE^−/−^ mice and investigated whether and to what degree NSA treatment alters the formation and progression of experimental AAAs via aortic morphological and histopathological analyses. The influences on abdominal aortic atherosclerosis, systemic levels of interleukin (IL)-1β and IL-18, lipid profiles, and blood pressure were also assessed.

## 2. Materials and Methods

### 2.1. AAA Induction and Intervention

Male ApoE^−/−^ mice at the age of 8 weeks were purchased from Nanjing Junke Bioengineering Ltd., Nanjing, Jiangsu, China, and used for all experiments. The use and care of animals were approved by Shanxi Medical University Animal Research Committee on Laboratory Animal Research (Taiyuan, Shanxi, China) in compliance with the Guidelines for the Ethic Review and Laboratory Animal Warfare, the People’s Republic of China National Standard GB/T35892-2018 [33].

AAAs were induced by subcutaneously infusing Ang II as previously reported [31,34,35,36,37]. Briefly, mice were anesthetized by 3% isoflurane inhalation, and a dorsal subcutaneous pocket was created. An ALZET osmotic minipump (Table 1) was implanted into the pocket to deliver human recombinant Ang II (1000 ng/min/kg body weight, Table 1) for 28 days. Three days prior to Ang II infusion, NSA (5 mg/kg body weight, Table 1) or an equal value of vehicle (50% PEG300 and 50% saline) was given to the mice daily via oral gavage for 30 days. NSA at 5 mg/kg body weight has been proven effective in suppressing GSDMD activity via the blockage of pore formation in vivo [27] (Figure 1).

### 2.2. Morphological Assessments and Definition of Experimental AAAs

Twenty-eight days following Ang II infusion, mice were euthanized via overdose inhalation of isoflurane, and aortas were harvested and digitally photographed with a ruler (Figure 1). Maximal adjacent infrarenal aortic external diameter, maximal suprarenal external aortic diameter, maximal suprarenal aortic external area, suprarenal aortic arch length, suprarenal aortic chord length, and maximal aortic area were measured using the NIH Image J software (Version 1.46r, https://imagej.nih.gov/ij/, accessed on 23 February 2024) (Figure 2). The ratios for maximal suprarenal external aortic diameter to maximal adjacent infrarenal aortic external diameter as well as suprarenal aortic arch length to suprarenal aortic chord length were also calculated. An AAA was defined as a more than 50% increase in maximal suprarenal external aortic diameter over the average baseline aortic diameter obtained from age-matched saline-infused male ApoE^−/−^ mice, the presence of aortic dissection (the presence of intramural hematoma), or mortality due to AAA rupture [34,37,38].

### 2.3. Measurements of Systolic and Diastolic Blood Pressure

Blood pressure was measured in restrained conscious mice placed on heating pads using a noninvasive tail-cuff monitoring system (Mode BP-2010A, Softron Biotechnology, Haidian, Beijing, China). For each mouse, five consecutive blood pressure readings were acquired and averaged. All measurements were conducted prior to (baseline level) and 28 days following Ang II infusion (Figure 1).

### 2.4. AAA Severity Grading

AAA severity was scored according to the presence or absence of aortic dilation and thrombus, as well as the shape and number of AAAs, as previously reported (Figure 1) [34,37,38]. AAA severity in the suprarenal aorta was scored as grade I to V: grade I: aortic dilation without thrombus; grade II: aortic dilation containing thrombus; grade III: pronounced bulbous form of aneurysm containing thrombus; grade IV: multiple aneurysms containing thrombus; and grade V: death due to aneurysm rupture.

### 2.5. Aortic Histological Analyses

Frozen suprarenal aortic sections (6 μm thickness) were prepared from mice 28 days following Ang II infusion and fixed with cold acetone for 8 min (Figure 1). Hematoxylin and eosin (H&E), elastic Verhoeff’s Van Gieson (EVG), and Masson trichome stains were performed to assess general aortic morphology and medial elastin and collagen deposition, respectively [39,40]. Aortic sections were immunostained with monoclonal antibodies (mAbs) against SMC α-actin, cluster of differentiation (CD) 68, CD4, CD8, and CD31 for SMCs, macrophages, CD4^+^ T cells, CD8^+^ T cells, and neoangiogenesis, separately. Medial elastin integrity was evaluated by counting the number of medial elastin breaks per aortic cross-section (ACS), while aortic wall collagen deposition and SMCs were quantified as positively stained areas as measured by Image J software. Macrophages, CD4^+^ T cells, CD8^+^ T cells and neoangiogenesis were presented as CD68^+^ cells, CD4^+^ cells, CD8^+^ cells and CD31^+^ blood vessels per ACS, respectively [39,40]. All reagents for histological analyses are listed in Table 1.

### 2.6. Staining and Quantification of Aortic Atherosclerotic Lesion

H&E and Oil Red O stains were performed on 10% formalin-fixed suprarenal aortic cross-sections to determine total and lipid atherosclerotic lesion sizes, respectively (Table 1 and Figure 1). Lesion macrophages were identified by CD68 mAb immunostaining. Lesion size and macrophages were quantitated using Image J software.

### 2.7. Measurements of Serum Cytokines and Lipids

Serum was prepared from the mice 28 days after Ang II infusion. The levels of IL-1β and IL-18 were quantified using commercial enzyme-linked immunosorbent assay (ELISA) kits (Table 1 and Figure 1) and expressed as pg/mL. Total cholesterol and triglyceride levels were also quantitated using commercial assay kits (Table 1) and presented as mg/dL.

### 2.8. Statistical Analyses

All statistical analyses were performed using GraphPad Prism (Version 9.3.1, GraphPad Software LLC, San Diego, CA, USA). The Shapiro–Wilk normality test was utilized to determine data normality. Data are presented as mean ± standard deviation (SDs) when normally distributed, or otherwise as median with the interquartile range (25% to 75%). Student’s *t* test or two-way analysis of variance followed by two-sample comparison was used to test statistical differences between groups for normally distributed data; otherwise, the non-parametric Mann–Whitney test was used. A *p* value less than 0.05 was considered statistically significant.

## 3. Results

### 3.1. NSA Treatment Alleviates Experimental AAA Enlargement

To determine the influence of NSA treatment on experimental AAAs, male ApoE^−/−^ mice were daily treated with NSA (5 mg/kg body weight) or an equal value of vehicle for 30 days beginning 3 days prior to Ang II infusion. Four weeks thereafter, mice were euthanized, and aortas were harvested for the morphological assessments of AAAs (Figure 2A). No difference was noted in adjacent external infrarenal aortic diameter between the NSA (mean ± SD: 0.9 ± 0.1 mm) and vehicle (0.8 ± 0.1 mm) groups (Figure 2B). The maximal suprarenal aortic external diameter was significantly smaller in NSA-treated mice (median and interquartile range: 1.2, 1.1–1.5 mm) than in vehicle-treated mice (2.4, 1.5–2.5 mm) (Figure 2C). Even after normalizing by the adjacent infrarenal aortic diameter, it remained statistically significant (Figure 2D). Though the length of the suprarenal aortic arch or chord was reduced in NSA- as compared to vehicle-treated mice, neither reached statistical significance (Figure 2E–G, 0.05 < *p* < 0.1). Consistent with the influence on suprarenal aortic diameter, the maximal suprarenal aortic area was also significantly reduced in NSA- (median and interquartile range: 9.9, 3.1–12.1 mm^2^) as compared to vehicle-treated mice (2.9, 2.4–4.2 mm^2^) (Figure 2H). Altogether, these results indicate that NSA treatment alleviated Ang II-induced experimental AAA enlargement.

### 3.2. NSA Treatment Reduces the Incidence and Lowers the Severity of Experimental AAAs

Next, we assessed the influence of NSA treatment on the incidence and severity of experimental AAAs. An AAA was defined by a more than 50% increase in maximal suprarenal external aortic diameter over the average baseline aortic diameter obtained from age-matched saline-infused male ApoE^−/−^ mice, the presence of aortic dissection (the presence of intramural hematoma), or mortality due to AAA rupture. AAA incidence was significantly lower in NSA- (4/11, 36.4%) than in vehicle-treated mice (9/12, 83.3%) (Figure 3A,B). Although NSA treatment tended to reduce AAA-specific mortality due to AAA rupture as compared to vehicle treatment, no difference was seen in the survival rate between the two treatment groups (Figure 3C, 0.05 < *p* < 0.1). Three vehicle-treated mice died due to AAA rupture, compared to no NSA-treated mice, as confirmed by autopsies.

We further evaluated the influence on AAA severity at sacrifice based on the presence and shape of aneurysmal thrombus as well as the number of AAAs, as reported previously [34,37,38]. As seen in Figure 3D, AAAs were scored as grade I in three mice, grade II in two mice, grade III in one mouse, grade IV in two mice, and grade V in three mice in the vehicle treatment group. In contrast, AAAs were scored as grade I in nine mice and grade II in two mice in the NSA treatment group. The median AAA severity score was significantly lower in NSA- (1.0, 1.0–1.0) than in vehicle-treated mice (2.5, 1.3–4.8) (Figure 3E). Thus, NSA treatment attenuated experimental AAA onset and severity.

### 3.3. NSA Treatment Reduces Medial Elastin Breaks, SMC Depletion, and Collagen Deposition in Experimental AAAs

At sacrifice, we assessed the influence of NSA treatment on characteristic experimental aneurysmal histopathologies, including medial elastin breaks, SMC depletion, and collagen deposition, on aortic frozen sections using EVG, SMC α-actin mAbs, and Masson trichome stains (Figure 4A). The number of medial elastin breaks was apparently reduced in NSA- (median and interquartile range: 7, 5–10/ACS) as compared to vehicle-treated mice (14, 12–16.5/ACS) (Figure 4B,C). Collagen deposition, as measured by collagen-positive area, was also reduced in NSA- (52,995, 39,426–134,810 μm^2^) as compared to vehicle-treated mice (151,653, 85,944–673,268 μm^2^) (Figure 4F–H). Additionally, SMC α-actin-positive SMC area was significantly increased in NSA- (80,306, 59,367–96,886 μm^2^) as compared to vehicle-treated mice (44,537, 37,064–62,361 μm^2^) (Figure 4D,E). These influences on elastin, collagen, and SMCs remained statistically significant even after normalizing with aortic internal perimeter on the cross-sections (Figure 4C,E,G). Together, these findings indicate that NSA treatment improved aortic structural integrity in experimental AAAs.

### 3.4. NSA Treatment Attenuates Leukocyte Accumulation and Angiogenesis in Aneurysmal Aorta

Leukocyte accumulation is the hallmark of AAA histopathology; thus, we performed immunostaining on frozen aortic sections for macrophages, CD4 T cells, and CD8 T cells using leukocyte subset-specific mAbs. Aortic macrophages, as identified by CD68 mAb staining, were significantly reduced in NSA- (59.3 ± 48.3 cells/ACS) as compared to vehicle-treated mice (170.2 ± 130.6 cells/ACS) (Figure 5A,B). Similarly, aortic CD4^+^ T cells and CD8^+^ T cells were also reduced in NSA- (median and interquartile range: 14, 8–29 cells/ACS for CD4^+^ T cells and 9, 4–21 cells/ACS for CD8^+^ T cells) as compared to vehicle-treated mice (47, 21–123 cells/ACS for CD4^+^ T cells and 25.15–57 cells/ACS for CD8^+^ T cells) (Figure 5A,C,D).

Because increased angiogenesis is another pathologic characteristic of AAAs, we assessed the influence of NSA treatment on aneurysmal angiogenesis using anti-CD31 mAb immunostaining (Figure 5E). We found dense CD31-positive neovessels in the aortas of vehicle-treated mice (median and interquartile range: 14, 6–51.5 vessels/ACS). Following NSA treatment, however, neovessel density was reduced to a median score of 3 with a 25–75% interquartile range of 0 and 10 vessels/ACS, which significantly differed from that in vehicle treatment. These results indicate that experimental AAA suppression by NSA treatment was associated with attenuated aortic accumulation of macrophages and T lymphocytes as well as reduced aneurysmal angiogenesis.

### 3.5. NSA Treatment Reduces Atherosclerosis in Suprarenal Aorta in ApoE^−/−^ Mice following Ang II Infusion

Ang II also promotes experimental atherosclerosis [41]; thus, we assessed whether NSA treatment suppresses atherosclerosis in suprarenal aortas in male ApoE^−/−^ mice following Ang II infusion. As represented in Figure 6A–C, H&E staining revealed a marked reduction in total atherosclerotic lesion area in NSA- as compared to vehicle-treated mice, even after normalized by aortic internal perimeter. Consistently, lipid lesion area, as measured by Oil Red O staining, was also significantly smaller in NSA- than in vehicle-treated mice, regardless of whether the area was normalized by total lesion size or aortic internal perimeter (Figure 6A,D,E). Additionally, NSA treatment alleviated macrophages in atherosclerotic lesions even after normalization by total lesion or aortic internal perimeter (Figure 6A,F,G,H). Thus, NSA treatment also ameliorated atherosclerosis in suprarenal aortas in Ang II-infused male ApoE^−/−^ mice.

### 3.6. NSA Treatment Reduces the Systemic Levels of IL-1β and IL-18 in Experimental AAAs

GSDMD mediates membrane pore formation, which promotes the secretion of IL-1β and IL-18, two proaneurysmal cytokines in AAA pathogenesis [9]. In our cytokine ELISA assays, serum IL-1β levels were 415.6 ± 250.3 pg/mL and 171.2 ± 36.8 pg/mL in vehicle and NSA treatments, respectively, representing a 58.8% reduction by NSA treatment (Figure 7A). Similarly, serum IL-18 levels were also significantly lower in NSA- (87.6 ± 27.6 pg/mL) than in vehicle-treated mice (199.7 ± 122.0 pg/mL) (Figure 7B). Thus, NSA treatment reduced the systemic levels of proaneurysmal IL-1β and IL-18.

### 3.7. NSA Treatment has no Impact on Body Weight Gain, Lipid Levels, and Blood Pressure in ApoE^−/−^ Mice following Ang II Infusion

Finally, we assessed whether AAA suppression by NSA is attributed to alterations in body weight, lipid levels, and/or blood pressure in ApoE^−/−^ mice following Ang II infusion. As shown in Figure 8A, body weight at either baseline level or 28 days following Ang II infusion did not differ between the two groups, indicative of no recognizable effect of NSA treatment on body weight gain. Although serum total cholesterol and triglyceride levels at sacrifice were lower in NSA- than in vehicle-treated mice, neither reached statistical significance (Figure 8B,C). Ang II infusion for 28 days increased both systolic and diastolic blood pressures, particularly systolic pressure, as compared to the corresponding baseline level. Again, no difference was seen for either systolic or diastolic pressure between the two treatment groups (Figure 8D,E). Thus, experimental AAA suppression by NSA treatment was not attributed to alterations in body weight, lipid levels, or blood pressure.

## 4. Discussion

In this study, we found that NSA treatment alleviated AAA enlargement and severity as well as lowered the incidence and mortality of AAAs. NSA treatment also ameliorated characteristic AAA pathologies, including medial elastin breaks, SMC depletion, collagen deposition, aortic leukocyte accumulation, and angiogenesis.

GSDMD is a pore-forming protein, which is cleaved to N- and C-terminals by active caspase-1 and caspase-4/5/11 generated by inflammasome activation or alternative sources [42,43,44,45]. GSDMD N-terminal oligomerizes to form cell membrane pores responsible for the release of mature IL-1β and IL-18 as well as pyroptosis [44,46,47,48]. It has been reported that GSDMD activation, as evidenced by elevated levels of its N-terminal, was markedly augmented in experimental and clinical aneurysmal aortas [24,25,26,49]. SMC-specific deletion of GSDMD attenuated experimental AAA progression with both Ang II infusion and porcine pancreatic elastase painting models in association with SMC phenotype switching [26]. In experimental and clinical aneurysmal aortas, macrophages expressed GSDMD, which not only prompted the secretion of inflammatory cytokines by macrophages but also caused pyroptosis in aortic SMCs [50]. The findings in the current study were consistent with our previous studies in which disulfiram, an alternative and functional distinct GSDMS inhibitor, attenuated Ang II-induced AAA formation and progression [37,51].

NSA was initially discovered as an inhibitor of mixed-lineage kinase domain-like pseudokinase, a pore-forming protein for necroptosis [52]. Recent work further identified NSA as a chemical GSDMD inhibitor, as it directly bound to GSDMD and blocked its N-terminal oligomerization [27]. It has been shown that NSA treatment suppressed both pyroptosis and necroptosis and thus post-resuscitation myocardial dysfunction [53]. NSA also promoted the proliferation and differentiation of osteoblasts through inhibiting NLRP3/caspase-1/GSDMD-mediated pyroptosis [54]. However, the influence of NSA on AAA has not been reported. In the present study, we found that NSA treatment mitigated Ang II-induced experimental AAAs in conjunction with reduced levels of serum IL-1β and IL-18. These findings support the hypothesis that the AAA suppression by NSA was in part mediated by inhibiting GSDMD activity. Our findings that NSA had no impact on body weight gain, lipid levels, or blood pressure preclude the possibility that obesity, hyperlipidemia, or high blood pressure may have remarkable contributions to NSA-mediated AAA inhibition.

Ang II also promoted experimental atherosclerosis [31]. In this study, we found that NSA treatment reduced total atherosclerotic lesions, lipid lesions, and macrophage accumulation in the suprarenal aorta. Thus, our study suggests that, in addition to experimental AAAs, NSA was also protective against atherosclerosis at least in the abdominal aorta in our experimental setting. Our findings are in accordance with previous studies showing that genetic deficiency of GASDM attenuated atherosclerosis in ApoE^−/−^ mice and high-fat-diet-fed low-density lipoprotein receptor-deficient mice [55,56]. Our study warrants further comprehensive investigation of the anti-atherosclerotic effect of pharmacological GSDMD inhibition in other aortic segments, such as the aortic root and arch.

The present study has several limitations. Due to technical limitations, we were unable to directly prove whether NSA treatment abrogates the oligomerization of cleaved GSDMD N-terminals in aortic aneurysmal lesions in vivo, as initially reported in in vitro observations [27]. NSA also reportedly inhibited necroptosis [52,57]. Pharmacologically inhibiting necroptosis with a small-molecule inhibitor (GSK2593074A) targeting receptor-interacting protein kinases 1 and 3 has been reported to attenuate experimental AAAs in an alternative AAA model [58]. Thus, we cannot rule out whether and to what extent the AAA suppression by NSA can be attributed to its effect on necroptosis. Additionally, our study was proof of the concept that pharmacologically targeting GSDMD inhibits AAAs. Investigating its influence on the progression of existing AAAs, dose–response, and the efficacy in alternative AAA models is beyond the scope of the present study.

In conclusion, NSA treatment ameliorated the formation and progression of Ang II-induced experimental AAAs in hyperlipidemic mice, with attenuated key aortic pathologies and systemic levels of IL-1β and IL-18. Thus, pharmacologically inhibiting GSDMD by NSA or alternative inhibitors may lead to the innovation of nonsurgical drug therapies for clinical AAA management.

## Figures and Tables

**Figure 1 biomolecules-14-00726-f001:**
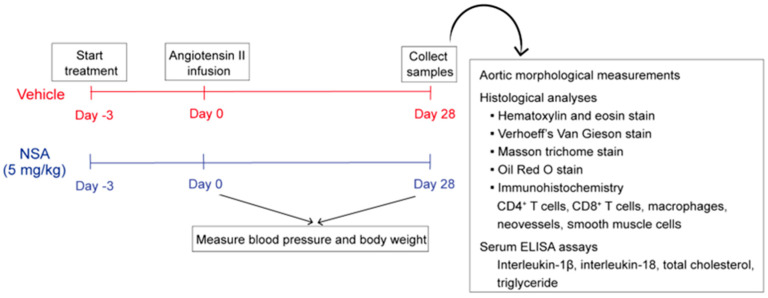
Study design and experimental approaches. Eight-week-old male apolipoprotein E-deficient mice received continuous subcutaneous infusion of angiotensin II (1000 ng/min/kg body weight) for 28 days to induce abdominal aortic aneurysms. Necrosulfonamide (NSA) (5 mg/kg body weight) or vehicle was given by daily oral gavage, beginning 3 days prior to angiotensin II infusion, for 30 days. Blood pressure measurements were performed prior to (day 0) and 28 days following angiotensin II infusion. Influences on abdominal aortic aneurysms were evaluated via aortic morphological measurements and histological analyses at sacrifice. Blood specimens were collected 28 days after angiotensin II infusion for the measurements of serum lipid and inflammatory cytokine levels. CD: cluster of differentiation. ELISA: enzyme-linked immunosorbent assay.

**Figure 2 biomolecules-14-00726-f002:**
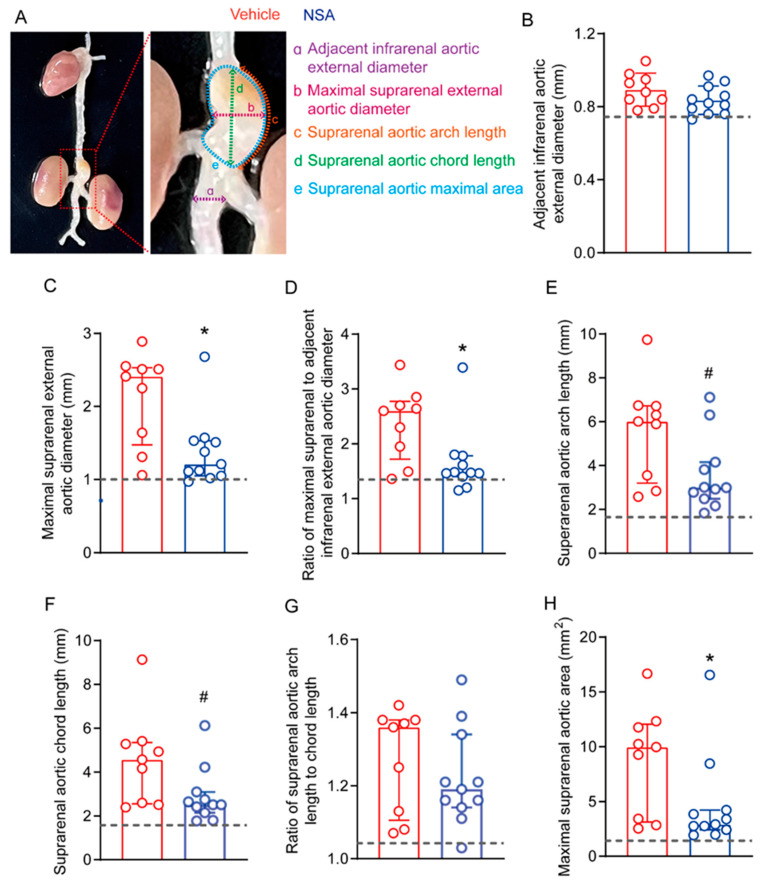
Necrosulfonamide (NSA) treatment attenuates experimental abdominal aortic aneurysm enlargement. Male apolipoprotein E-deficient mice were subcutaneously infused with angiotensin II (1000 ng/min/kg body weight) for abdominal aortic aneurysm induction. Mice were treated with vehicle or NSA (5 mg/kg body weight) for 30 days, initiating 3 days prior to angiotensin II infusion. (**A**) Scheme for morphological evaluation of angiotensin II-induced abdominal aortic aneurysms. (**B**) Mean ± standard deviation of adjacent infrarenal external aortic diameter. (**C**–**H**) Median and interquartile range (25% and 75%) of maximal suprarenal external aortic diameter (**C**), ratio of maximal suprarenal to adjacent infrarenal external aortic diameter (**D**), suprarenal aortic arch length (**E**), suprarenal aortic chord length (**F**), ratio of suprarenal arch length to chord length (**G**) and maximal suprarenal aortic area (**H**). Non-parametric Mann–Whitney, 0.05 < # *p* < 0.1 and * *p* < 0.05 compared to vehicle treatment. Dotted lines: the average value for each morphological parameter measurement from age-matched saline-infused apolipoprotein E-deficient mice.

**Figure 3 biomolecules-14-00726-f003:**
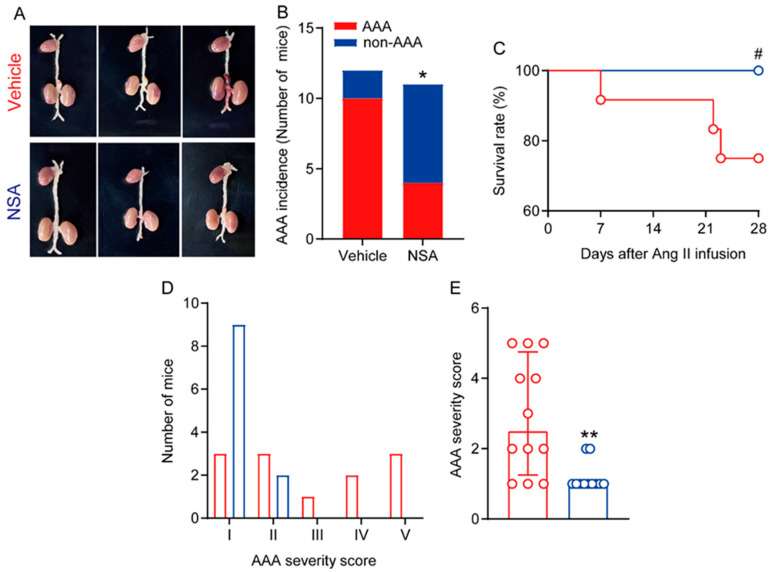
Necrosulfonamide (NSA) treatment attenuates the formation and severity of experimental abdominal aortic aneurysms (AAAs). Male apolipoprotein E-deficient mice were infused with angiotensin (Ang) II (1000 ng/min/kg body weight) for 28 days to induce AAAs. (**A**) Representative abdominal aortic aneurysm images in the vehicle and NSA treatment groups. (**B**) AAA incidence. AAA was defined by at least a 50% increase in suprarenal aortic diameter over that in the age-matched saline-infused mice, the presence of aortic dissection, or death caused by aneurysm rupture. Fisher’s exact test, * *p* < 0.05 compared to vehicle treatment. (**C**) Survival rate. Log-rank test, 0.05 < # *p* < 0.1 compared to vehicle treatment. (**D**) Distribution of AAA severity score. AAAs were graded as grade I to V based on the presence of intramural thrombus as well as the shape and number of aneurysms. (**E**) Quantification of AAA severity score (median and interquartile range (25% and 75%)) in two treatment groups. Non-parametric Mann–Whitney, ** *p* < 0.01 compared to vehicle treatment.

**Figure 4 biomolecules-14-00726-f004:**
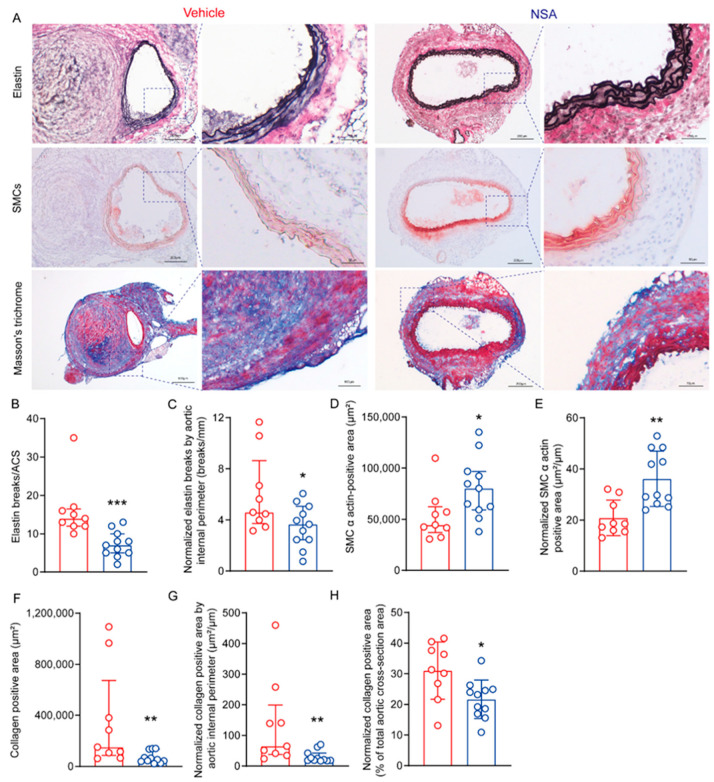
Necrosulfonamide (NSA) treatment reduces medial elastin breaks, smooth muscle cell (SMC) depletion, and collagen deposition in experimental abdominal aortic aneurysms. Apolipoprotein E-deficient mice were sacrificed 28 days following angiotensin II infusion. Aortas were harvested, sectioned (8 μm), and stained via the elastic Verhoeff’s Van Gieson staining for medial elastin, SMC α-actin antibody for SMCs, and Masson trichome staining for collagen deposition. (**A**) Representative histological staining images for elastin (black to blue/black), SMCs (red), and collagens (blue). (**B**,**C**) Quantification of medial elastin breaks. (**D**,**E**) Quantification of SMC α-actin-positive area in aortic cross-sections (ACSs). (**F**–**H**) Quantification of collagen-positive area in ACSs. All data in (**C**,**E**,**G**,**H**) are normalized by aortic internal perimeter in ACSs. Student’s *t* test ((**D**,**F**): mean ± standard deviation) and non-parametric Mann–Whitney ((**B**,**C**,**G**,**H**): median and interquartile range (25% and 75%)), * *p* <0.05, ** *p* < 0.01 and *** *p* < 0.001 compared to vehicle treatment.

**Figure 5 biomolecules-14-00726-f005:**
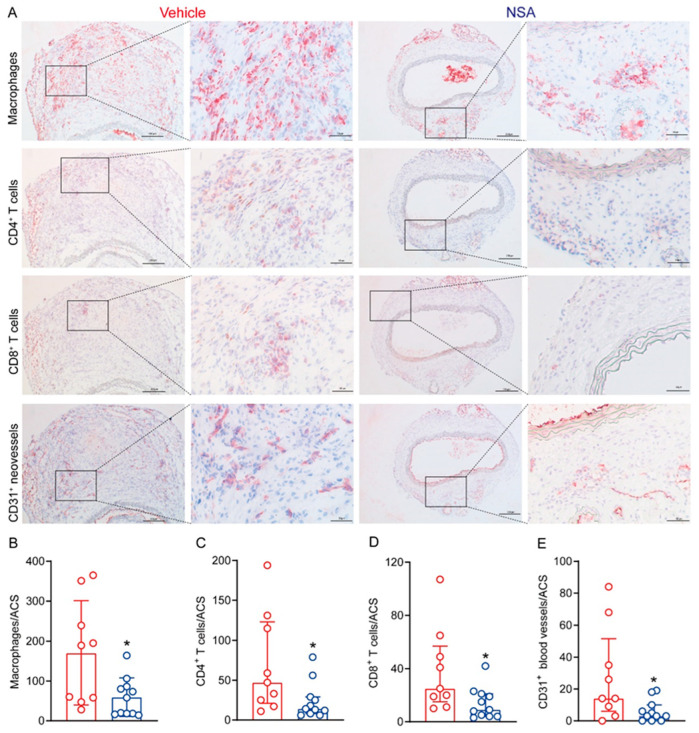
Necrosulfonamide (NSA) treatment reduces aortic leukocyte accumulation and angiogenesis in experimental abdominal aortic aneurysms. Frozen aortic sections from vehicle- and NSA (5 mg/kg body weight)-treated, angiotensin II-infused mice were fixed with acetone and stained with antibodies against CD68 for macrophages, CD4 for CD4^+^ T cells, CD8 for CD8^+^ T cells, CD31 for neovessels. (**A**) Representative immunohistochemical staining images for macrophages, CD4^+^ T cells, CD8^+^ T cells, and neovessels in two treatment groups. (**B**) Quantification (mean ± standard deviation) of macrophage accumulation. (**C**–**E**) Quantification (median and interquartile range) of CD4^+^ T cells (**C**), CD8^+^ T cells (**D**), and CD31^+^ neovessels (**E**) per aortic cross-section (ACS). Student’s *t* test (**B**) and non-parametric Mann–Whitney (**C**–**E**), * *p* < 0.05 compared to vehicle treatment. CD: Cluster of differentiation.

**Figure 6 biomolecules-14-00726-f006:**
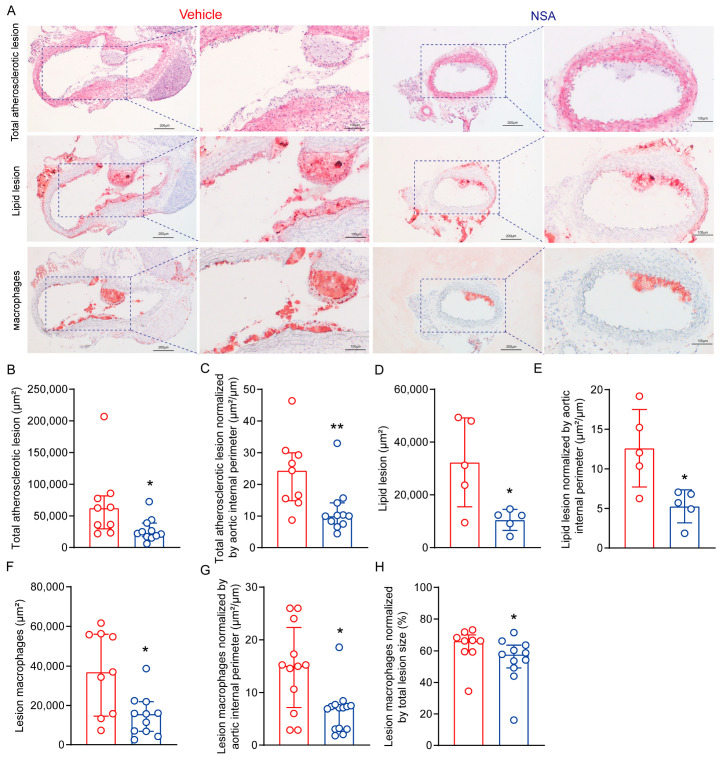
Necrosulfonamide (NSA) treatment reduces atherosclerotic lesion size and alleviates macrophage accumulation in male apolipoprotein E-deficient mice following angiotensin II infusion. Frozen aortic sections were prepared from differentially treated angiotensin II-infused apolipoprotein E-deficient mice, fixed with 10% formalin or acetone, and stained with hematoxylin and eosin (acetone-fixed sections) for total atherosclerotic lesions, Oil Red O for lipid lesions (10% formalin-fixed sections), and CD68 for macrophage accumulation (acetone-fixed sections). (**A**) Representative histological images of total lesions (hematoxylin and eosin stain), lipid lesions (Oil Red O stain), and macrophage accumulation (CD68 monoclonal antibody immunostaining). (**B**,**C**) Quantification (median and interquartile range) of total atherosclerotic lesions (**B**) and total atherosclerotic lesions normalized by aortic internal perimeter (**C**). (**D**,**E**) Quantification (mean ± standard deviation) of lipid deposition (**D**) and lipid deposition normalized by aortic internal perimeter (**E**). (**F**–**H**) Quantification (median and interquartile range) of macrophage accumulation (**F**), macrophage accumulation normalized by total lesion size (**G**), and macrophage accumulation normalized by aortic internal perimeter (**H**). Student’s *t* test (**D**,**E**) and non-parametric Mann–Whitney (**B**,**C**,**F**–**H**), * *p* < 0.05 and ** *p* < 0.01 compared to vehicle treatment.

**Figure 7 biomolecules-14-00726-f007:**
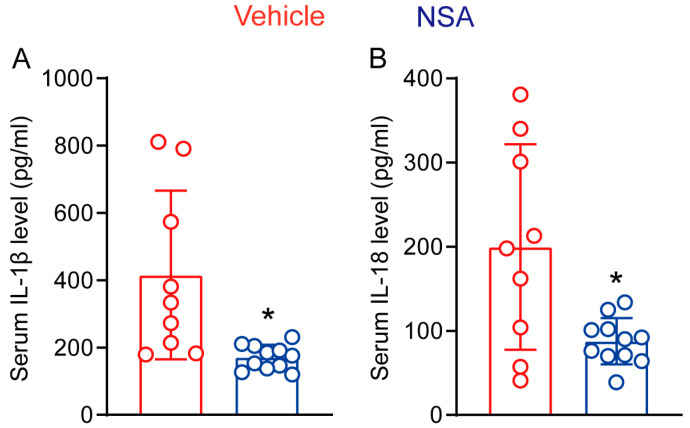
Necrosulfonamide (NSA) treatment reduces the serum levels of interleukin (IL)-1β and IL-18 in experimental abdominal aortic aneurysms. Twenty-eight days after angiotensin II infusion, sera were prepared from differentially treated apolipoprotein E-deficient mice. Serum levels of IL-1β (**A**) and IL-18 (**B**) were assessed via enzyme-linked immunosorbent assay. All data are mean ± standard deviation. Student’s *t* test, * *p* < 0.05 compared to vehicle treatment.

**Figure 8 biomolecules-14-00726-f008:**
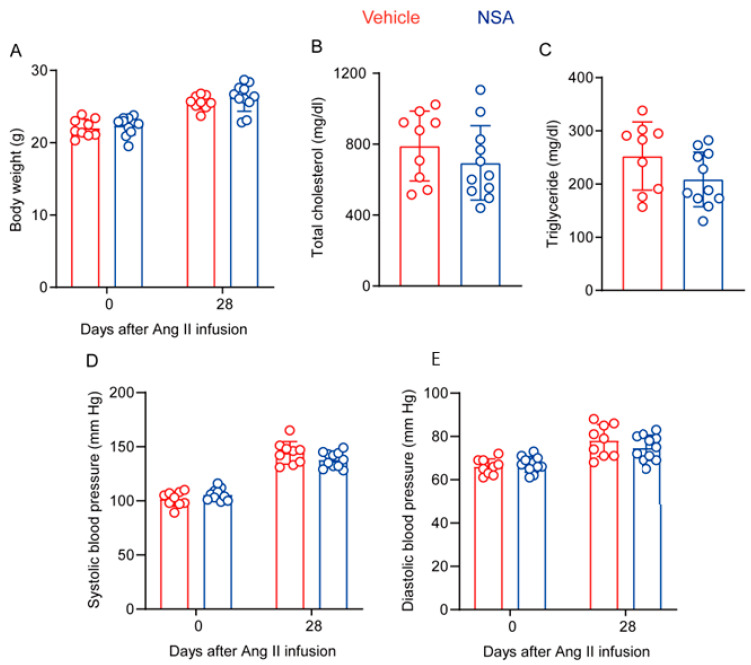
Necrosulfonamide (NSA) treatment does not affect body weight gain, lipid levels, or blood pressure in experimental abdominal aortic aneurysms. (**A**) Body weight in vehicle- and NSA-treated apolipoprotein E-deficient mice at the baseline (day 0) and 28 days after angiotensin (Ang) II infusion. (**B**,**C**) Serum levels of total cholesterol (**B**) and triglycerides (**C**) from vehicle- and NSA-treated apolipoprotein E-deficient mice 28 days after Ang II infusion. (**D**,**E**) Systolic (**D**) and diastolic (**E**) blood pressure of vehicle- and NSA-treated apolipoprotein E-deficient mice at the baseline and 28 days after Ang II infusion. All data are presented as the mean ± standard deviation. Two-way analysis of variance test (**A**,**D**,**E**) or Student’s *t* test (**B**,**C**) showed no significant difference between the two treatment groups at the same timepoint.

**Table 1 biomolecules-14-00726-t001:** Major reagents and materials.

Material/Reagent	Maker	Catalog Number	Clone Number	Working Solution Concentration or Dose
Apolipoprotein E-deficient mice	Nanjing Junke Bioengineering Ltd.	N/A	N/A	N/A
Human angiotensin II	MedChemExpress	HY-13948	N/A	1000 ng/min/kg
Alzet mini-osmotic pump	Durect Corportaion	2004	N/A	N/A
Necrosulfonamide	MedChemExpress	HY-100573	N/A	5 mg/kg
PEG300	MedChemExpress	HY-Y0873	N/A	N/A
Hematoxylin staining solution	Biosharp	BL702B	N/A	N/A
Eosin staining solution	Solarbio Science and Technology Co.	G1100	N/A	N/A
Elastic Verhöeff Van Gieson stain kit	Leagene Biotechnology Co.	DC0059	N/A	N/A
Masson’s trichrome stain kit	Solarbio Science and Technology Co.	G1340	N/A	N/A
Oil Red O	Solarbio Science and Technology Co.	O8010	N/A	5 mg/mL
Mayer’s Hematoxylin stain solution	Solarbio Science and Technology Co.	G1080	N/A	N/A
Mouse IL-1β ELISA kit	Solarbio Science and Technology Co.	SEKM-0002	N/A	N/A
Mouse IL-18 ELISA kit	Solarbio Science and Technology Co.	SEKM-0019	N/A	N/A
Total cholesterol assay kit	Nanjing jiancheng Bioengineering Institute	A111-1-1	N/A	N/A
Triglycerides assay kit	Nanjing jiancheng Bioengineering Institute	A110-1-1	N/A	N/A
Rat anti-mouse CD4 mAb	Biolegend Inc.	100402	GK 1.5	2.5 μg/mL
Rat anti-mouse CD8 mAb	Biolegend Inc.	100702	53–6.7	2.5 μg/mL
Rat anti-mouse CD31 mAb	Biolegend Inc.	102402	390	5.0 μg/mL
Rat anti-mouse CD68 mAb	Biolegend Inc.	137002	FA-11	2.5 μg/mL
Horseradish peroxidase-conjugated mouse anti-mouse smooth muscle cell α-actin mAb	Santa Cruz Biotechnology Co.	sc-32251 HRP	1A4	4.0 μg/mL
Biotinylated rabbit anti-rat IgG antibody	Boster Biological Technology Co.	BA1005	N/A	5.0 μg/mL
Streptavidin–peroxidase conjugate	Solarbio Science and Technology Co.	SE068	N/A	5.0 μg/mL
AEC peroxidase substrate Kit	Sigma-Aldrich Crop.	AEC101	N/A	N/A

AEC: 3-amino-9-ethylcarbazole; CD: cluster of differentiation; ELISA: enzyme-linked immunosorbent assay; IL: interleukin; mAb: monoclonal antibody; PEG: polyethylene glycol.

## Data Availability

All data are included within this article.

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
