# Peer review of "Gasdermin D Inhibitor Necrosulfonamide Alleviates Angiotensin II-Induced Abdominal Aortic Aneurysms in Apolipoprotein E-Deficient Mice"

_biomolecules, 2024, doi:10.3390/biom14060726_

Round 1
Reviewer 1 Report
Comments and Suggestions for Authors
The authors explored the pharmacological inhibition of NSA and its potential impact on AAAs. Using a mouse model of AAA induced by AngII in hyperlipidemic mice, the authors demonstrated that NSA, an inhibitor of gasdermin D, effectively mitigated both the formation and progression of AAA. This was evidenced by a reduction in key aortic pathologies and systemic levels of IL-1b and IL-18. NSA treatment resulted in the attenuation of aneurysm formation and progression, characterized by decreased infiltration of inflammatory cells and preserved medial elastin and smooth muscle cells density. Overall, the manuscript is well-written and provide valuable insights into the potential therapeutic role of NSA in managing AAAs.
Minor points:
1) The statement “AAA incidence was significantly lower in NSA- (9/12, 83.3%), than that in vehicle- (4/11, 36.4%), treated mice (Figs. 3A & 3B)” seems not correct.
2) Figure 3 legend should be corrected to “twenty-eight” days.
3) In Figure 3C. it appears that none of NSA treated mice were died by day 28. The statement that no difference was observed in survival rates between the two treatment groups seems contradictory and requires clarification.
4) Please ensure consistency by using “aortae” or “aortas” uniformly throughout the text.
5) In the Figure 6 legend, please remove the phrase “figure 6”.
Comments on the Quality of English Languagenone
Author Response
Thank you very much for your reviewing our manuscript and comments.

Reviewer 2 Report
Comments and Suggestions for Authors
Major points
At a first glance, this study on NSA and AAA appears coherent and carefully conducted. Unfortunately, the main issue lies with the histochemical and immunohistochemical stains, which are captured at too low magnification and do not appear to be representative. For instance, this reviewer cannot identify any specific staining for SMC in Fig. 1 A. Additionally, an image of the Masson's trichrome stain, which is used to assess collagen content, is missing. A similar problem exists with Figure 6: Large parts of the adventitia are stained and apparently included in the evaluation, which is not representative. These issues are critical, as the entire evaluation depends on valid histochemical and immunohistochemical staining.
Minor points
What is the proposed mechanism of the induction of AAA by Ang II? This should be stated in the introduction.
Page 7: … NSA- (9/12, 83,3 %), than that in vehicle- (4/11, 36,4 %) should be the other way round (see parentheses)
Comments on the Quality of English LanguageEnglish language needs improvement:
Page 2: „with the later“ = with the latter
Table 1: „Huaman“ = human
Figure 2: „Superarenal“ = suprarenal
Figure 3: „twenty-dight“ = twenty-eight
Author Response
Many and many thanks to your time for reviewing our manuscript and wonderful suggestions. All these greatly improved our manuscript quality.
